# Inhibition of polymerase chain reaction: Pathogen-specific controls are better than human gene amplification

**Guillaume Roux[1][¤], Christophe Ravel[2], Emmanuelle Varlet-Marie[2], Rachel Jendrowiak[2], Patrick Bastien[2], Yvon Sterkers[2]\***

**1** Univ. Montpellier, and Laboratoire de Microbiologie, Centre Hospitalier Universitaire (CHU) of Nîmes, Nîmes, France, **2** Univ. Montpellier, Centre Hospitalier Universitaire (CHU) of Montpellier, Dept. of Parasitology-Mycology, Research Unit MiVEGEC, CNRS, IRD, Montpellier, France

¤ Current address: Centre Hospitalier de Gap, Laboratoire de Biologie, Gap, France
* yvon.sterkers@umontpellier.fr

**Data Availability Statement:** All relevant data are within the paper and its Supporting Information files.

## Abstract

PCR inhibition is frequent in medical microbiology routine practice and may lead to false-negative results; however there is no consensus on how to detect it. Pathogen-specific and human gene amplifications are widely used to detect PCR inhibition. We aimed at comparing the value of PCR inhibitor detection using these two methods. We analysed Cp shifts (ΔCp) obtained from qPCRs targeting either the albumin gene or the pathogen-specific sequence used in two laboratory-developed microbiological qPCR assays. 3152 samples including various matrixes were included. Pathogen-specific amplification and albumin qPCR identified 62/3152 samples (2.0%), and 409/3152 (13.0%) samples, respectively, as inhibited. Only 16 samples were detected using both methods. In addition, the use of the Youden's index failed to determine adequate Cp thresholds for albumin qPCR, even when we distinguished among the different sample matrixes. qPCR targeting the albumin gene therefore appears not adequate to identify the presence of PCR inhibitors in microbiological PCR assays. Our data may be extrapolated to other heterologous targets and should discourage their use to assess the presence of PCR inhibition in microbiological PCR assays.

## Introduction

Molecular biology and particularly real-time PCR (qPCR) has revolutionized the biological diagnosis of infectious diseases. Nonetheless, PCR may fail because it is based on an enzymatic reaction susceptible to various mechanisms of inhibition [1]. Inhibition of PCR reaction is frequent in clinical microbiology and exposes to the risk of false negative results, hence PCR inhibition screening is recommended [2, 3]. PCR inhibition appears as a hardly predictable event and data about its actual frequency in routine practice of clinical biology laboratories are scarce [4, 5]. Differential susceptibility of each type of qPCR to different inhibitors and heterogeneity of sample matrixes make its detection non trivial. Many methods to detect PCR inhibition have been reported and some guidelines have been published. The use of PCR controls with a defined quantity of DNA molecules to check for the presence of a Cp switch due to the

PCR inhibition

**Funding:** "RSI Assurance Maladie Professions Libérales - Provinces. C.A.M.P.L.P." has paid the LightCycler 480 (Roche®) real-time PCR equipment in 2009. The funders had no role in study design, data collection and analysis, decision to publish, or preparation of the manuscript.

**Competing interests:** We acknowledge the financial support of the "RSI Assurance Maladie Professions Libérales - Provinces. C.A.M.P.L.P." for buying the LightCycler 480 (Roche®) real-time PCR equipment. Since this funder had no role in study design, data collection and analysis, decision to publish, or preparation of the manuscript; this does not alter our adherence to PLOS ONE policies on sharing data and materials.

presence of inhibitors in the sample extract is relevant; however it is risky since technicians may have to manipulate and distribute target DNA, thus increasing the risk of inter-well contamination. 'Internal' amplification controls, based on alien DNA added at a low concentration in the specimen before DNA extraction, and of which the presence must be checked in a duplex PCR together with the target, is also highly relevant. Most of commercial inhibition controls are based on this principle. The rationale of using a human gene as an extraction or inhibition control is less acceptable because the human DNA target is present in high quantity in the sample as compared to the target DNA [6]. Importantly, there is presently no consensus for PCR inhibition detection in routine practices. For example, in the single field of molecular diagnosis of toxoplasmosis in France, seven different methods are used to detect PCR inhibition in 30 laboratories [7]. Most of them use either a pathogen pathogen-specific PCR or a human gene amplification method. The former is an internal control made of either genomic *T. gondii* DNA or a plasmid containing the targeted pathogen DNA sequence. It has one major drawback which is the increased risk of false positive results due to increased amounts of amplicons and airborne contamination of reaction wells. Amplification controls may be prepared from genomic DNA (as in our *Toxoplasma*-PCR assay) or from the target DNA sequence cloned in a plasmid (as in our *Pneumocystis*-PCR assay). One development of the plasmid strategy is to clone a chimeric sequence to be amplified by the same primers than the PCR target but detected by specific probe(s) [8–11]. In the second type of method, albumin, beta-globin or human RNase P genes are targeted. This method is found attractive since (i) they can be implemented in all qPCR assays involving human samples; and (ii) they constitute a complete process control for DNA extraction and amplification. Yet, human gene qPCR cycle of positivity (Cp) depends also on the initial human DNA content or cellularity in the clinical sample, which is highly variable. Indeed, DNA content or cellularity depends on the size/volume of the sample, the matrix, *i.e.* the nature of the sample, such as blood or cerebrospinal fluid, and on the pathophysiological state of the patients. The control of DNA extraction is another critical step of molecular diagnosis, but was not explored in our study. In addition, it is reported in the literature that PCR methods differing by their primers and/or amplified sequence have variable susceptibility to inhibitors [12–14]. These studies tend to invalidate the assumption that absence of inhibition in a qPCR targeting any human gene has a good predictive value for assessing inhibition in a qPCR targeting a pathogen. Consequently, it is critical to assess the performances of human gene-based PCR inhibition screening methods in clinical samples in clinical microbiology routine practice.

In this study, (i) we analyse the frequency of PCR inhibition in a large range of clinical samples and (ii) we show that human gene-based qPCR methods are not efficient to affirm the presence/absence of PCR inhibitors.

## Methods

We retrospectively analysed *Toxoplasma* and *Pneumocystis* qPCR tests for all the clinical samples analysed in 2016 in the Department of Parasitology-Mycology at the academic hospital of Montpellier (Montpellier, France).

### DNA extraction

The DNA extraction method was dependent on the PCR assay and the sample matrix. For *Pneumocystis*-PCR, all types of respiratory samples were analysed, including bronchoalveolar lavage fluid (BALF), sputum, bronchoaspiration, nasal aspiration and pleural fluid; DNA was extracted using QIAamp DNA minikit (Qiagen) according to the manufacturer specifications. For *Toxoplasma*-PCR, the DNA extraction method depended on the matrix. For low cellular

samples such as cerebrospinal fluids (CSF), aqueous humor, amniotic fluids and crystal clear BALF, DNA was extracted using the Tween-Nonidet-NaOH method[15]. When *Pneumocystis* and *Toxoplasma* qPCRs were performed on the same sample, QIAamp DNA minikit (Qiagen) was used. Other samples were processed using protein precipitation solution (A795A; Promega, France)[16]. As recommended, DNA extracts were then stored at– 20˚C prior to PCR for best preservation[17].

### *Toxoplasma* and *Pneumocystis* qPCR assays

The *Toxoplasma* qPCR targeted the 'rep529' sequence (AF146527) and was performed as described by Reischl et al. [18]. The *Pneumocystis* qPCR targeted the major surface glycoprotein gene (AF372980) and was performed as described by Fillaux et al.[19]. We used LC480® Probe Master 2X (Roche®) for both qPCR and Uracil-N-Glycosylase (Roche®) for *Toxoplasma* qPCR. Each PCR well contained 15 µL of mix and 5 µL of DNA extract for *Toxoplasma* qPCR and 18 µL of mix and 2 µL of DNA extract for *Pneumocystis* qPCR. Amplification and detection occurred in LightCycler® 480 (Roche) and raw fluorescence data were analysed using the LightCycler® 480 software release 1.5.0 (Roche). Cycles of positivity were determined by the second derivative method.

### PCR controls and definition of inhibition

Each PCR plate contained one well for negative control (sterile water) and one for positive control (calibrated positive sample). Cp values of positive plate controls were plotted into Levey-Jennings charts and interpreted according to the Westgard rules[20]. DNA extraction control was performed in routine practice by qPCR targeting the human albumin gene[21]. Amplification controls were made of reaction wells containing pathogen DNA on top of the patient DNA extract. For the *Toxoplasma* qPCR assay, this pathogen-specific amplification positive control was prepared from a $10^5$ *T. gondii* tachyzoites/mL freeze-dried standard as described in Varlet-Marie et al. [22]; 2 µL of *T. gondii* DNA extract were added to obtain a final concentration of 1.5 *T. gondii* genome/tube for which the expected Cp value is 35.2 ±1.5. The control was performed in duplicate. A sample was considered inhibited if both amplification control wells showed a Cp>38. If only one well showed a Cp>38, we considered this as irrelevant and due to stochastic events. For *Pneumocystis*-qPCR, the amplification control was made of the pathogen target DNA sequence, which had been cloned into pGEM-T easy®. Plasmid DNA maxipreparation (Qiagen Maxiprep®) was then diluted in order to reach a Cp of 30.1 ±1.2, and 1 µL of this dilution was added to the reaction tube. A *Pneumocystis*-qPCR reaction was considered inhibited if the Cp difference between amplification control and the positive plate control was ≥3. In this study, albumin-qPCR was considered as inhibited if the difference between the Cp of albumin-qPCR ($Cp_{alb}$) obtained for a sample and the mean $Cp_{alb}$ for the matrix considered was ≥3.

### Data management

PCR results and bio-clinical data were exported from LightCycler® 480 release 1.5.0 software (Roche®) and DxLab® software (MedaSys®) respectively, to spreadsheets (S1 and S2 Figs).

### Determination of the PCR efficiencies

PCR efficiencies of the techniques were determined using a standard curve generated by performing a logarithmic serial dilution of DNA extracts. The standard curves demonstrating a linear relationship between the logarithm of the copy number and the Cp value, allows to determine the PCR efficiency using: Efficiency = -1+10exp(-1/slope) [23].

## Statistics

Data were analyzed using R 3.4.0 and RStudio 1.0.143 softwares [24, 25]. We calculated the Youden's index considering the pathogen-specific amplification control as the reference method to detect PCR inhibition. The Youden's index combines sensitivity and specificity into a single measure (Sensitivity + Specificity– 1) and its value ranges between -1 and 1. A value of 1 indicates that there are neither false positive nor false negative results. The best $Cp_{alb}$ cut-off values of albumin-qPCR to detect inhibition were defined using the Youden's index. Sensitivities and specificities of these $Cp_{alb}$ cut-offs were next compared by binomial tests to the 95% theoretical value in order to assess their inferiority. Correlation of $Cp_{alb}$ to leucocyte counts was assessed using Spearman's rank correlation. Holm correction was performed for all the tests. Samples without $Cp_{alb}$ value due to a flat amplification curve were excluded from the graph and the correlation. A p value $<0.05$ was considered as significant.

## Ethical approval and informed consent

This work was carried out in accordance with the relevant guidelines and regulations; it does not include potentially identifying patient/participant information. The study corresponds to a non-interventional retrospective study and according to the French Health Public Law (CSP Art L1121-1.1), such studies are exempt from informed consent requirement and do not require approval by an ethics committee.

## Results and discussion

### General description

The Dept. of Parasitology-Mycology is a regional and national reference centre which coordinates the "Molecular biology pole" of the National Reference Center for Toxoplasmosis (http://cnrtoxoplasmose.chu-reims.fr). The laboratory is accredited according to ISO 15189:2012 for the *Toxoplasma* and *Pneumocystis* PCR assays. The *Toxoplasma*-qPCR assay is being used in routine since July 2009, >17000 clinical samples have been routinely tested, and PCR efficiency is regularly controlled and found at ≥97.5% [18]. For the *Pneumocystis*-qPCR assay, the corresponding features are March 2013, >2500, and 90 ±9%. PCR efficiency of the albumin qPCR was 98.5% (Fig 1). These laboratory-developed qPCR assays may therefore be considered as robust and finely optimized.

During the year 2016, 3152 samples were referred to the laboratory as part of the routine activity (Fig 2): 2225 samples were analysed by *Toxoplasma*-qPCR, 964 by *Pneumocystis*-qPCR, and 37 samples were analysed by both methods (S1 Table). Two samples analysed by *Toxoplasma*-qPCR were excluded from this study, one due to cancellation of the analysis and another due to identity problem (S1 and S2 Figs). Most *Toxoplasma*-qPCR analysed samples were blood samples (1874 out of 2225), followed by 95 CSF and 114 postnatal samples (Table 1). No *Pneumocystis*-qPCR samples had to be excluded from the analysis; the 964 *Pneumocystis*-qPCR samples comprised 787 BALF and 141 sputa samples (Table 1). The percentage of positive samples for the *Toxoplasma*-qPCR was 1.7%, ranging from 0.3% for the blood to 12% for the amniotic fluid samples. For *Pneumocystis*-qPCR, the percentage of positive sample was 7.1% (Table 1).

### Inhibition rates in clinical samples as assessed from the pathogen-specific amplification control

Of the 3152 analysed samples, 62 (2.0%) showed evidence of PCR inhibition as assessed from the pathogen-specific amplification control: 45 samples for *Toxoplasma*-qPCR and 17 for

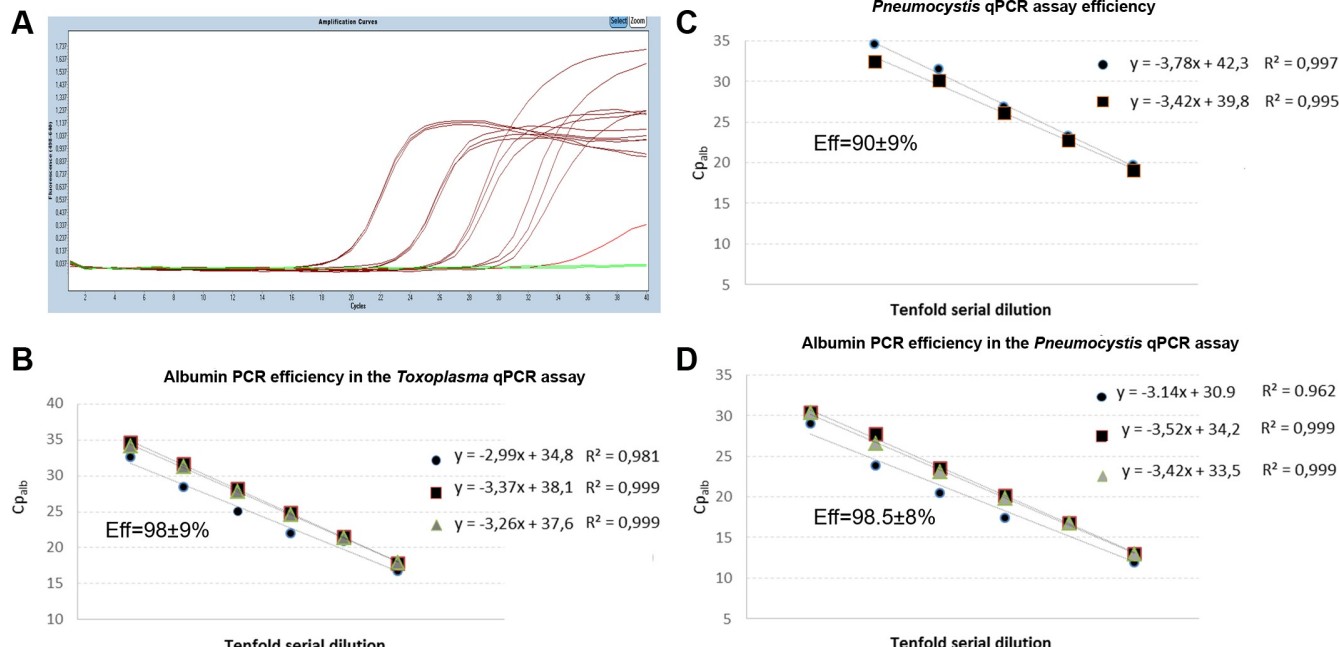

**Fig 1. PCR efficiencies. A**. LightCycler real-time PCR amplification curves in a serial dilution assay. **B-D**. PCR efficiencies were determined in triplicate in two/three independent experiments. PCRs were performed in triplicate on five/six samples representing a 10-fold serial dilution. All standard curves demonstrated a linear relationship between the logarithm of the copy number and the Cp value, allowing determining the PCR Efficiency using: Efficiency = -1+10exp(-1/slope).

*Pneumocystis*-qPCR (Fig 2). However, this percentage varied from 1.8–15% depending on the matrix (Table 1). In 61/62 inhibited samples, tenfold dilution of the DNA extract released inhibition, confirming that the unexpected Cp values were due to PCR inhibitors. The remaining sample was a cord blood tested for the presence of *Toxoplasma* for which PCR remained

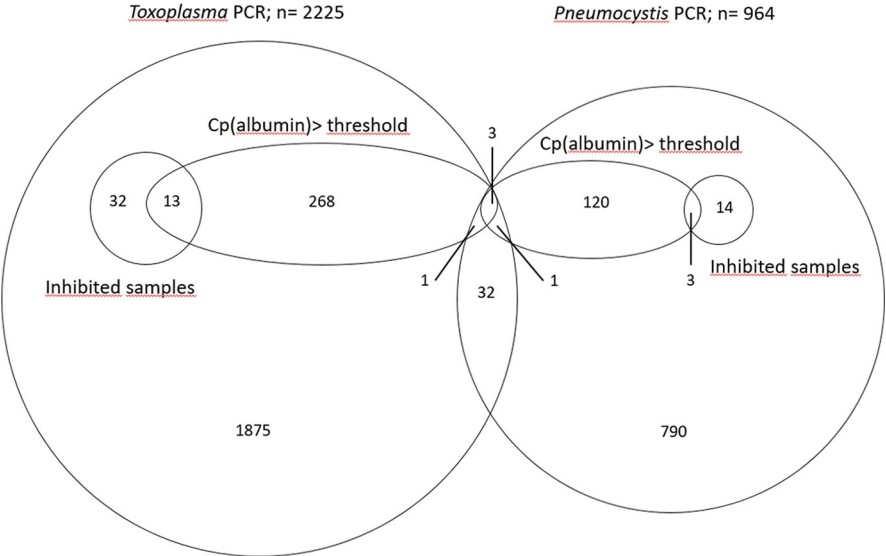

**Fig 2. Venn diagram of number of analyzed samples.** Inhibited samples: Samples were considered inhibited if the Cp of the pathogen-specific control was >38 for both wells for *Toxoplasma*-PCR (N = 45), and if ΔCp ≥3 for *Pneumocystis*-PCR (N = 17). Cp(albumin)>threshold: For albumin-PCR, the inhibition threshold was the mean $Cp_{alb}$ + 3 for a given matrix (see Methods and Table 1).

**Table 1. Frequency of PCR inhibition as a function of sample matrix for *Toxoplasma* PCR and *Pneumocystis* PCR.**

| | | *Toxoplasma* PCR | | | | | | | *Pneumocystis* PCR | | | | | | |
|---|---|---|---|---|---|---|---|---|---|---|---|---|---|---|---|
| | | Positive samples | | Inhibited samples | | $Cp_{alb} \geq$ threshold* | | | Positive samples | | Inhibited samples | | $Cp_{alb} \geq$ threshold* | | |
| | | Nber | % | Nber | % | Threshold | Nber | % | Nber | % | Nber | % | Threshold | Nber | % |
| Blood | | 4/1309 | 0.3 | 31/1309 | 2,3 | 23,1 | 104/1309 | 7,9 | - | - | - | - | - | - | - |
| Blood from leucopenic patient | | 7/565 | 1.2 | 2/565 | 0,4 | 27,6 | 123/565 | 21,8 | - | - | - | - | - | - | - |
| CSF | | 1/95 | 1.1 | 3/95 | 3.2 | 34,9 | 16/95 | 16,8 | - | - | - | - | - | - | - |
| Placenta | | 3/58 | 5.2 | 4/58 | 6.9 | 21,7 | 7/58 | 12,1 | - | - | - | - | - | - | - |
| Cord blood | | 2/56 | 3.6 | 2/56 | 3.6 | 23,8 | 10/56 | 17,9 | - | - | - | - | - | - | - |
| AF | | 15/44 | 34.1 | 0/44 | 0 | 29,3 | 5/44 | 11,4 | - | - | - | - | - | - | - |
| BALF** | | 1/41 | 2.4 | 0/41 | 0 | 29.9 | 5/41 | 12.2 | 51/787 | 6,5 | 12/787 | 1.5 | 24.7 | 100/787 | 12.7 |
| Aqueous humor | | 2/28 | 7.1 | 0/28 | 0 | 33,6 | 8/28 | 28,6 | - | - | - | - | - | - | - |
| Sputum | | - | - | - | - | - | - | - | 16/141 | 11.4 | 4/141 | 2.8 | 23,9 | 23/141 | 16,3 |
| Bronchoaspiration | | - | - | - | - | - | - | - | 1/35 | 2.9 | 1/35 | 2.8 | 22,1 | 4/35 | 11,4 |
| Miscellaneous** | | 1/29 | 3.4 | 3/29 | 10,3 | 32 | 7/29 | 24,1 | 0/1 | 0 | 0/1 | 0 | 24.7 | 0/1 | 0 |
| | Cerebral biopsy | 1/14 | 7.1 | 2/14 | 14.3 | 28,4 | 1/14 | 7.1 | - | - | - | - | - | - | - |
| | Lymph node | 0/6 | 0 | 0/6 | 0 | 28,4 | 1/6 | 16.7 | - | - | - | - | - | - | - |
| | Liver biopsy | 0/2 | 0 | 1/2 | 50 | 28,4 | 1/2 | 50 | - | - | - | - | - | - | - |
| | Ascites | 0/2 | 0 | 0/2 | 0 | 28,4 | 1/2 | 50 | - | - | - | - | - | - | - |
| | Pericardia fluid | 0/2 | 0 | 0/2 | 0 | 28,4 | 1/2 | 50 | - | - | - | - | - | - | - |
| | Pleural fluid | 0/1 | 0 | 0/1 | 0 | 28,4 | 1/1 | 100 | - | - | - | - | - | - | - |
| | Bone marrow | 0/1 | 0 | 0/1 | 0 | 28,4 | 0/1 | 0 | - | - | - | - | - | - | - |
| | Retina | 0/1 | 0 | 0/1 | 0 | 28,4 | 1/1 | 100 | - | - | - | - | - | - | - |
| | Nasal aspiration | - | - | - | - | - | - | - | 0/1 | 0 | 0/1 | - | 24,7 | 0/1 | 0 |
| Total | | 37/2225 | 1,7 | 45/2225 | 2 | | 285/2225 | 12,8 | 68/964 | 7,1 | 17/964 | 1.8 | | 127/964 | 13,2 |

CSF: cerebrospinal fluid; AF: amniotic fluid (including external quality assessment samples); BALF: bronchoalveolar lavage fluid. Samples were considered inhibited if Cp of amplification controls was >38 for both amplification control wells for *Toxoplasma* PCR, and if ΔCp was ≥3 for *Pneumocystis* PCR

*threshold = mean Cp + 3

** for *Toxoplasma* and *Pneumocystis* PCR respectively, 37 BALF samples were analysed by both methods (for details, see Methods).

inhibited despite the dilution, showing flat amplification curves for both the target- and the albumin-specific inhibition controls before and after tenfold dilution. No additional dilution was applied to this sample due to the risk of reduced sensitivity. Interestingly, blood samples, which are reported elsewhere [26] to be prone to PCR inhibition did not exhibit the highest inhibition rate in our study. Samples like sputum or placenta were also prone to PCR inhibition, probably because it is difficult to optimize a DNA extraction protocol for heterogeneous matrixes with highly different structures and properties from one sample to another. Using a similar pathogen-specific control method to detect PCR inhibitors in BALF samples, Döskaya *et al.* found a 23.8% inhibition rate for *Pneumocystis* PCR [5] but a large study of Buckwalter *et al.* reported inhibition rates under 1% on 386,706 samples [4]. These discrepant results illustrate that PCR inhibition measurement is a problem of variable importance in routine practices, probably depending on the ability of extraction methods to remove inhibitors, on the susceptibility of PCR methods to inhibitors and on the performance of PCR inhibition detection.

## Performances of PCR inhibition detection using albumin-qPCR

409/3152 samples (12.9%) exhibited a $Cp_{alb}$ above the threshold, *i.e.* above the mean Cp for the matrix concerned + 3 (Table 1). Thus, (i) the albumin-qPCR identified many more samples as

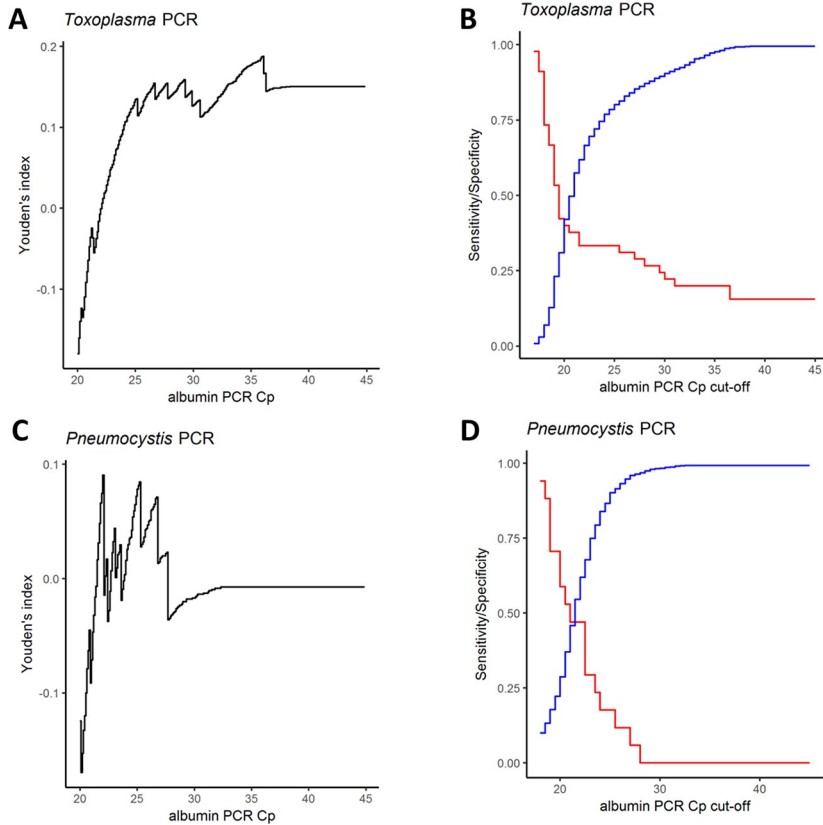

**Fig 3. Youden's index, sensitivity and specificity of albumin-PCR cut-off values. A and C**. Black curves: Youden's index = sensitivity + specificity– 1; **B and D**. Red curves: sensitivity; blue curves: specificity. For *Toxoplasma* qPCR **(A)**, the maximal Youden's index was very low. The maximum value was 0.19 and was reached for a $Cp_{alb}$ cut-off value of 36. At this albumin qPCR cut-off value sensitivity is 20% (p< 0.0001; Binomial test for 95% theoretical sensitivity) and specificity 98.8% (p = 1, Binomial test for 95% theoretical specificity). **(B)** Graphical analysis of sensitivity and specificity curves in function of cut-off values. Both curves intersect at a $Cp_{alb}$ value of 22. At this $Cp_{alb}$ cut-off value sensitivity is 33.3% and specificity 66.6% (p<0.0001 and p<0.0001; Binomial tests for 95% theoretical sensitivity and specificity). For *Pneumocystis* qPCR **(C)**, the maximal Youden's index was too low to allow determining any $Cp_{alb}$ cut-off value. **(D)** Graphical profiles of sensitivity and curves in function of cut-off values. Both curves intersect at a $Cp_{alb}$ value of 22. At this $Cp_{alb}$ cut-off value sensitivity is 47.1% and specificity 62% (p<0.0001 and p<0.0001; Binomial tests for 95% theoretical sensitivity and specificity).

containing inhibitors than the pathogen-specific controls, and this was so in both qPCR assays, *i.e.* 285 *vs.* 45 for the *Toxoplasma* assay, and 127 *vs.* 17 for the *Pneumocystis* assay. In addition, (ii) it is noteworthy that only 13 and 3 samples were detected as inhibited using both inhibitor-detection methods, for the *Toxoplasma* and *Pneumocystis* assays, respectively (Fig 2). We therefore conclude that (i) the human gene-based qPCR detected inhibitors much more often than the pathogen-specific qPCR and (ii) it most often did not detect them correctly. Considering this poor correlation between pathogen-specific controls and albumin qPCR to detect PCR inhibitors, we wished to determine $Cp_{alb}$ thresholds more adapted to our assays. To do so, we calculated the sensitivity, specificity of the albumin qPCR to detect PCR-inhibiting samples using the pathogen-specific control as reference and the Youden's index (Sensitivity + Specificity– 1) and we varied the $Cp_{alb}$ threshold (see Methods and Fig 3A). We could not determine a $Cp_{alb}$ threshold allowing to obtain 95% sensitivity and 95% specificity (Fig 3B), whether this strategy was used for all matrixes together or for each type of matrix (S3 and S4 Figs). We therefore concluded that no $Cp_{alb}$ cut-off values can be determined to efficiently detect pathogen-specific inhibitions for both pathogen-specific PCR assays. To check the

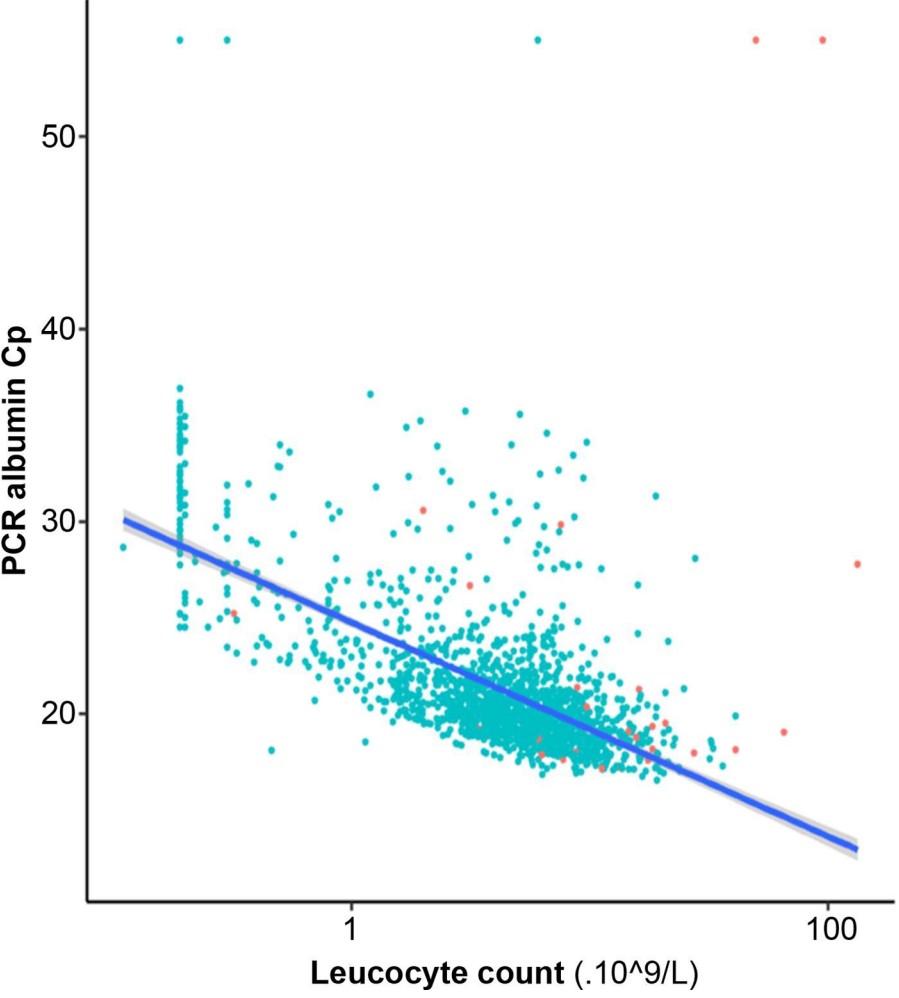

**Fig 4. Correlation between leucocyte counts and Cp$_{alb}$ values in 1690 blood samples.** Semi-log scale graph where leucocyte count (x-axis) was plotted against Cp$_{alb}$ (orange and green dots corresponding to amplification control inhibited and not inhibited samples respectively). Linear model was used to determine trend line (Blue) with 95% confidence interval (Grey).

influence of the cellularity of clinical samples on Cp$_{alb}$ values, we compared white blood cell counts to albumin qPCR results for blood samples. We analysed 1690/1874 blood samples; the remaining 184 samples were excluded from the analysis due to the absence of whole blood cell count on the same day as *Toxoplasma*-PCR. Cp$_{alb}$ values were correlated to leucocyte counts (p<0.0001; Spearman's rank correlation), but for a given white blood cell count a large range of Cp$_{alb}$ was observed (Fig 4). The lack of sensitivity and specificity of albumin PCR to detect target specific inhibitions may be explained by a differential susceptibility of each qPCR assay to inhibitors and by the variable quantity of human DNA in clinical samples. These results prevent using human gene-based PCR, *e.g.* albumin, betaglobin or human RNase P genes, as PCR inhibition detection method. We based our demonstration on two models, *i.e. Toxoplasma* and *Pneumocystis* but these results should be expanded to the detection of other pathogens in human samples.

Another widely used method to search for inhibitors is the use of commercial "universal" controls. These commercial controls are made of exogenous DNA and are added either in the

sample before extraction or with the DNA extract in the PCR mix. Differential susceptibility to inhibitors and efficiency discrepancies between PCR assays should also prove problematic in this approach. Indeed, the size and GC rate of the amplicons of the foreign DNA used will have an impact on the detection of the inhibition [13]. So, implementation of one of these controls should be avoided until their performances have been assessed in routine practice.

In conclusion, pathogen-specific amplification controls appear to be a method of choice for screening the presence of inhibitors in a PCR assay for infectious diseases as compared to the use of a human gene-based qPCR.

## Supporting information

**S1 Table. Spreadsheet of raw data.**
(XLSX)

**S1 Fig. Flow chart of data cleaning and tidying of *Toxoplasma* PCR results.**
(DOCX)

**S2 Fig. Flow chart of data cleaning and tidying of *Pneumocystis* PCR results.**
(DOCX)

**S3 Fig. Youden's index, sensitivity and specificity of albumin PCR cut-off values for the most frequent matrixes analysed in *Toxoplasma* PCR.**
(DOCX)

**S4 Fig. Youden's index, sensitivity and specificity of albumin PCR cut-off values for the most frequent matrixes analysed in *Pneumocystis* PCR.**
(DOCX)

## Acknowledgments

We are grateful to S. Douzou, B. Sanichanh and G. Serres for their technical help.

## Author Contributions

**Conceptualization:** Yvon Sterkers.

**Data curation:** Guillaume Roux.

**Formal analysis:** Guillaume Roux, Christophe Ravel, Yvon Sterkers.

**Funding acquisition:** Patrick Bastien, Yvon Sterkers.

**Investigation:** Guillaume Roux, Rachel Jendrowiak.

**Methodology:** Yvon Sterkers.

**Project administration:** Yvon Sterkers.

**Resources:** Emmanuelle Varlet-Marie, Rachel Jendrowiak, Yvon Sterkers.

**Supervision:** Yvon Sterkers.

**Validation:** Yvon Sterkers.

**Visualization:** Yvon Sterkers.

**Writing – original draft:** Guillaume Roux, Yvon Sterkers.

**Writing – review & editing:** Christophe Ravel, Patrick Bastien, Yvon Sterkers.

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
