## [Decision Letter · Decision Letter 0]

26 Jul 2019

PONE-D-19-16663

Original research. Inhibition of Polymerase Chain Reaction: Pathogen-Specific Controls are better than Human Gene Amplification

PLOS ONE

Dear Dr Sterkers,

Thank you for submitting your manuscript to PLOS ONE. After careful consideration, we feel that it has merit but does not fully meet PLOS ONE’s publication criteria as it currently stands. Therefore, we invite you to submit a revised version of the manuscript that addresses the points raised during the review process.

The paper is definetely worthy, but I feel that the statistics evaluation could be improved.

Follow the suggestions given by the referee and check the quality of your images.

We would appreciate receiving your revised manuscript by August, 14th. To enhance the reproducibility of your results, we recommend that if applicable you deposit your laboratory protocols in protocols.io, where a protocol can be assigned its own identifier (DOI) such that it can be cited independently in the future. For instructions see: http://journals.plos.org/plosone/s/submission-guidelines#loc-laboratory-protocols

We look forward to receiving your revised manuscript.

Kind regards,

Antonella Marangoni, Ph.D.

Academic Editor

PLOS ONE

""RSI Assurance Maladie Professions Libérales - Provinces. C.A.M.P.L.P." has paid the LightCycler 480 (Roche®) real-time PCR equipment in 2009

We note that you received funding from a commercial source: RSI Assurance Maladie Professions Libérales - Provinces. C.A.M.P.L.P.

Please respond by return email with your amended Competing Interests Statement and we will change the online submission form on your behalf.

Reviewers' comments:

Reviewer's Responses to Questions

**Comments to the Author**

1. Is the manuscript technically sound, and do the data support the conclusions?

Reviewer #1: Yes

2. Has the statistical analysis been performed appropriately and rigorously? 

Reviewer #1: Yes

3. Have the authors made all data underlying the findings in their manuscript fully available?

Reviewer #1: No

4. Is the manuscript presented in an intelligible fashion and written in standard English?

Reviewer #1: Yes

5. Review Comments to the Author

Reviewer #1: Authors present their work on inhibition of PCR. I found this work useful. Following are my few suggestions in general and few minor edits for the manuscript, that authors may consider.

Overall, it’ll be a good to provide a brief discussion of the work in the context of it’s utility to other type of pathogens as well. In addition, it’ll be helpful for others if authors could provide the analysis pipeline on GitHub or something similar. Since they used R, therefore it should be easy. Authors may also improve the quality of figures as I found them bit blurred.

Minor comments

Line 140: I think it should be pathogen- specific amplification.

Line 148-150: Is the p-value significant computed after correction or before?

Line 194: should be pathogen-specific control. Also, please check space issues elsewhere in the article.

6. PLOS authors have the option to publish the peer review history of their article (what does this mean?). If published, this will include your full peer review and any attached files.

Reviewer #1: No

---

## [Author Response · Author response to Decision Letter 0]

16 Aug 2019

We were able to addresses all the points raised during the review process, see the uploaded response to reviewer file

---

## [Decision Letter · Decision Letter 1]

26 Aug 2019

[EXSCINDED]

Original research. Inhibition of Polymerase Chain Reaction: Pathogen-Specific Controls are better than Human Gene Amplification

PONE-D-19-16663R1

Dear Dr. Sterkers,

We are pleased to inform you that your manuscript has been judged scientifically suitable for publication and will be formally accepted for publication once it complies with all outstanding technical requirements.

With kind regards,

Antonella Marangoni, Ph.D.

Academic Editor

PLOS ONE

Additional Editor Comments (optional):

Reviewers' comments:

Reviewer's Responses to Questions

**Comments to the Author**

1. If the authors have adequately addressed your comments raised in a previous round of review and you feel that this manuscript is now acceptable for publication, you may indicate that here to bypass the “Comments to the Author” section, enter your conflict of interest statement in the “Confidential to Editor” section, and submit your "Accept" recommendation.

Reviewer #1: All comments have been addressed

2. Is the manuscript technically sound, and do the data support the conclusions?

Reviewer #1: Yes

3. Has the statistical analysis been performed appropriately and rigorously? 

Reviewer #1: Yes

4. Have the authors made all data underlying the findings in their manuscript fully available?

Reviewer #1: Yes

5. Is the manuscript presented in an intelligible fashion and written in standard English?

Reviewer #1: Yes

6. Review Comments to the Author

Reviewer #1: (No Response)

7. PLOS authors have the option to publish the peer review history of their article (what does this mean?). If published, this will include your full peer review and any attached files.

Reviewer #1: No

---

## [Editor Report · Acceptance letter]

20 Sep 2019

PONE-D-19-16663R1 

Inhibition of Polymerase Chain Reaction: Pathogen-Specific Controls are better than Human Gene Amplification. 

Dear Dr. Sterkers:

I am pleased to inform you that your manuscript has been deemed suitable for publication in PLOS ONE. Congratulations! Your manuscript is now with our production department. 

With kind regards,

on behalf of

PhD Antonella Marangoni 

Academic Editor

PLOS ONE